# Epidemics and local governments in struggling nations: COVID-19 in Lebanon

Jida M. Al-Mulki[1]*, Mahmoud H. Hassoun[2], Salim M. Adib[1]

**1** Department of Epidemiology and Population Health, American University of Beirut, Beirut, Lebanon,
**2** Department of Critical Care, Rafik Hariri University Hospital, Beirut, Lebanon

* jma54@mail.aub.edu

**Data Availability Statement:** All results files are available from the Mendeley Database (DOI: 10.17632/k4jp3xscg2.1).

**Funding:** The authors received no specific funding for this work.

## Abstract

Municipalities in Lebanon represent local governments at the basic community level. The proximity of the municipality to the local community and its knowledge of available resources, can be crucial in easing the impact of any disaster. This study aimed to document the range of preparedness/reactivity of municipalities as COVID-19 swept through Lebanon. A qualitative case study was implemented to explore municipal response to control the epidemic, using in-depth semi-structured interviews with twenty-seven stakeholders from nine municipalities across all governorates in Lebanon. In each municipality, participants included mayors/deputy mayors, available members of municipal councils, prominent community leaders, health care professionals, and managers of local NGOs. The collected data were analyzed using the comparative thematic analysis. The socioecological model was adopted to illustrate the dynamic interplay between the barriers and facilitators at all ecological levels. The response to the pandemic differed significantly in volume and nature among different municipalities across regions, with rural areas clearly disadvantaged in terms of adequacy and completeness of response. Barriers consistently mentioned by most municipalities included *economic collapse and poverty*, *shortage in resources*, *lack of support from the central government*, *stigma*, *lack of awareness*, *underreporting*, *flaws in the MOPH surveillance system*, *impeded accessibility to healthcare services*, *limited number and weak role of municipal police*, *increased mental illnesses*, and *political patronage, favoritism, and interference*. On the other hand, *increased donations*, *community engagement*, *social support and empathy*, *sufficient human resources*, *the effective role of healthcare systems*, and *good governance* were identified as key facilitators. The socioecological model identified several multi-level facilitators and loopholes which can be addressed through a suggested strategic "roadmap" providing evidence-based interventions for future epidemics. It is crucial meanwhile that the central government strengthens the administrative and financial resources of municipalities in preparing and rapidly deploying the expected optimal response.

**Competing interests:** The authors have declared that no competing interests exist.

## A. Introduction

### A1. Background

**Municipalities and service delivery in Lebanon during the COVID-19 pandemic.** The COVID-19 pandemic which started in China in the Fall of 2019, officially reached Lebanon in the second half of February 2020, and spread out across all regions, entailing large social, economic, and psychological consequences. The cumulative caseload reached 538,218 cases and 7,670 deaths by May 17,2021 in a country with an estimated population of 5 million. At the time of writing this paper in June 2021, the epidemic appeared to be in full recession in the country, and travelers from Lebanon had been classified as "green" in accessing the European Union space (S1 Fig). The epidemic hit Lebanon at a time of unprecedented financial and political crisis, devaluation of the local currency and rapid erosion of the middle-class [1].

Municipalities in Lebanon represent a form of governance which connects the central government with the population. Municipalities are autonomous entities with a wide array of financial and administrative local authority. Each community votes a municipal council ("majles baladiyeh") headed by a mayor for a period of six years. Municipalities are requested to fulfill numerous mandates and services at the local level such as street cleaning, public lighting, wastewater treatment, and other developmental activities [2]. Municipalities are under the official supervision of the Ministry of Interior and Municipalities (MI&M). Since the beginning of the COVID-19 crisis in Lebanon, the municipal councils have been entrusted with several tasks beyond their traditional services, to halt the community spread of the virus. Measures have included raising awareness, adopting "track, isolate, test, and treat" strategy, facilitating social distancing, enforcing home quarantine, regulating opening/closing times for businesses, assisting the community (financial aid, providing food, free vouchers. . .), along with a myriad of other ad-hoc services [3, 4]. Minimal functions mandated by MI&M are detailed in S1 File.

The effectiveness of the efforts varied a lot across municipalities based on the territorial dimension and therefore on the financial taxation capacity, revenue diversion to a shattered and broke central government, shortage in local staff, poverty, and passive resistance from the community [5]. For instance, in Tripoli, the largest city in Northern Lebanon, the population did not abide with the lockdown measures despite the governmental decision of "general mobilization" in March 2020. This was stated by the head of Tripoli's municipality who attributed community resistance to the prevailing poverty, with more than 40% of the population below the poverty line, and the limited financial capacities of the city. On the other hand, other municipalities had voluntarily taken distinctive initiatives beyond the government's decisions such as, launching entertainment activities to motivate people to stay home [6]. Thus, the impact of this epidemic affected communities differently even in the same governorate, due to the diversity of preparedness and resources across towns and neighborhoods.

**Municipalities in the pandemic: A worldwide review.** Several reports worldwide highlighted the major role which local governments, such as municipalities or regional councils, can have on the local expression of the epidemic outside large urban communities. The control of epidemics is a multi-sectoral activity which brings together health and non-health stakeholders, all working together to prevent the spread of the infection and to mitigate its impact. The proximity of the municipality to the local community and its knowledge of available resources can be crucial in easing the impact of any disaster. In Kosovo, municipalities engaged their citizens in the decision-making process throughout the pandemic, setting the ground for more local resilience [7]. The local government of Taipei, the capital city of Taiwan, undertook economic relief measures to prevent the collapse of local businesses and to alleviate unemployment rates. These measures included tax deferrals, reduction in rental fees and

many other subsidies. Similarly, several municipalities in Mexico implemented "Mercomuna", an initiative to facilitate community access to food items in urban markets, by providing special vouchers to families and small businesses to support them during the crisis [8].

## A2. Problem statement

In the absence of clear and well-rehearsed guidelines, municipalities' response in case of epidemics in Lebanon appeared to range from the felt presence to the virtually absent. A qualitative survey was conducted with several municipalities from different areas of Lebanon as the local epidemic was receding, to explore municipal decision-making processes regarding the local containment of COVID-19, to describe the levels of preparedness and reactivity, and to create a strategic "road map" with practical implications to optimize the response in future similar situations.

**Socio-ecological model: A guiding framework.** Our survey used the "socio-ecological model" (SEM) to define facilitators and barriers which may enable or impede municipal decision-makers to successfully contain the storm. SEM is an evidence-based framework that guides population-wide interventions through studying the dynamic interplay between multiple levels of influence [9]. SEM has been adapted in multiple contexts in Asia, Africa, and Near East to tackle different health outcomes. For instance, this framework has been used to analyze the response of higher organizations in preventing and controlling Ebola epidemic in Liberia, depicting how these barriers overlap and interact at the individual, community, service delivery, and policy levels (9). The ultimate aim of this framework is to target mechanisms of change through developing systematic interventions at different levels of influence.

# B. Methodology

## B1. Research design

A qualitative case study was implemented using in-depth-semi-structured interviews with different municipalities to explore their perceptions regarding the facilitators and barriers that impacted the response of municipalities during COVID-19 pandemic to guide multi-faceted public health interventions.

## B2. Sampling strategy

"Purposive sampling" was used to select a diversified list of municipalities, taking in account geographical, socio-cultural, and demographic characteristics as well as perceived activity profiles of the municipality during the pandemic. Municipalities finally included in our study were those that were subjectively deemed as "easily accessible" using all possible communication means in a period where direct contacts were not always allowed. Phone numbers of mayors of selected municipalities were obtained from (MI&M) electronic pages publicly available. Three key informants with relevant position and contribution during the pandemic were selected as sources of data in each municipality to ensure data triangulation. The recruitment process of stakeholders in each municipality is explained thoroughly in S2–S5 Files.

## B3. Sources of data

Key informants were selected in each municipality is based on a decision tree detailed in S3–S5 Files. The following categories of key informants were targeted in each community, in this priority order:

1. Mayors "president of the municipality" or their deputies.

2. Available members of the Municipal Council.

3. Relevant community leaders not currently members of the municipal council: health care professionals, lawyers involved in civil society. . .

4. Managers of NGOs active in that specific community.

### B4. Study procedures

Stakeholders initially contacted were interviewed, using phones or distance computer calls. Interviewees described the initiatives taken by the municipalities at varied levels to mitigate the impact of the pandemic on their communities. Initial respondents were queried for "significant" other local stakeholder, thus creating a "snowball" effect to canvass all the point of views. Interviews were conducted during the second wave of the pandemic, particularly between November 2020 and January 2021. On average, interviews typically took between 30 and 40 minutes. Interviews were audio-recorded and transcribed after getting approval from the interviewees.

### B5. Plan of analysis

Themes and subthemes were generated from transcripts. Constant comparative analysis was conducted to explore those themes and identify recurrent issues, and data saturation principle was adopted. Themes pertaining to the "facilitators" and "barriers" were compiled from all interviewed municipalities. Based on the identified initiatives taken, municipalities were divided into two categories, municipalities with "adequate response", and those with "partial or inadequate response".

- Municipalities that showed adequate response included those with active municipal members who took personal initiatives in collaboration with other key actors in the community.

- Municipalities with inadequate or partial response included those which were completely absent and did not take initiatives beyond the basic mandates, or those with inactive municipal members but where the civil society volunteers overtook the role of the municipality and bridged the gap with their own resources.

To keep this paper concise, the typical experience of two municipalities from each of the two categories described above will be the main focus throughout this study.

### B6. Ethical considerations

This study was approved by the Institutional Review Board (IRB) at the American University of Beirut (AUB). Oral informed consent was obtained from all participants, following a clear explanation of the study objectives, since the study was carried out remotely and a written consent was not feasible at that time. The oral consent along with the name of participants and date of interviews were documented through recordings. This consent procedure was advised and approved by the IRB committee. Potential participants were informed that names will not be disclosed in any report. All persons contacted were freely invited to participate and practically none of those contacted refused to participate. Interviews were audio-taped with participants' approval, and all transcripts were deleted after completion of the process.

## C. Results

In total, 27 in-depth interviews were carried out in nine municipalities from all governorates in Lebanon. As depicted in Fig 1, two municipalities were selected from North governorate

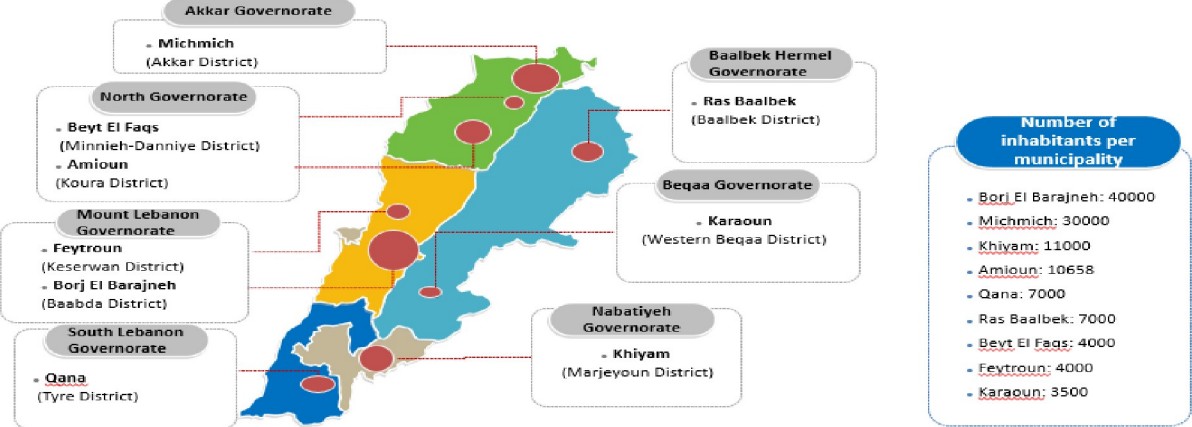

**Fig 1. Distribution of selected municipalities per governorates and population density.** Reprinted from [www.yourfreetemlates.com] under a CC BY-ND 4.0, with permission from the available website, original copyright [2017].

(*Beyt El-Faqs and Amioun*), one from Akkar governorate *(Michmich)*, two from Mount Lebanon governorate *(Borj El Barajneh and Feytroun)*, one from Beqaa governorate *(Karaoun)*, one from Baalbek-Hermel governorate *(Ras Baalbek)*, and two from South Governorate *(Qana and Khiyam)*.

The facilitators, barriers, and initiaives taken by each of the above municipalities were reported. However, for the rest of this analysis, and as mentioned earlier, we will detail results from two municipalities with "adequate" response: "Khiyam" and "Amioun" and two with "inadequate or partial" response: "Michmich" and "Beyt El Faqs".

## C1. Response of municipalities

Michmich and Beyt El Faqs municipalities undertook no more than the basic MI&M mandates. While the response in Michmich was totally absent, in Beyt El Faqs it was compensated by civil society volunteers who performed measures such as detecting and managing cases at home, providing medications for free, securing ambulances, and covering hospital admissions to needy people. Conversely, Khiyam and Amioun municipalities, with adequate response, were distinguished with a myriad of diverse initiatives. Aside from the basic mandates, they were able to detect cases, trace contacts, provide home care management, free of charge PCR testing, hospital coverage and free provision of COVID-19 and other chronic medications. Additionally, Khiyam municipality provided uninterrupted power supply and Internet services to all the town, at a time when such public services were in shortage and had to be purchased from private providers. Those municipalities provided psychological and occupational services to families locked down by the disease, and Amioun municipality even purchased vaccines to short-circuit tedious procurement through central government and expedite the immunization process in the community. Details of initiatives reported in the four municipalities are shown in Tables 1 & 2, respectively.

## C2. Facilitators

Seven facilitators were identified among the selected municipalities including: *Increased donations from well-off citizens and emigrants, Engagement of diversified community actors, Sufficient human resources, Good governance*, and *employing unique skills, Heightened sense of social support and empathy*, and *Effective role of healthcare systems*, (Table 3) (Fig 2).

**Table 1. Response of the active and partially active/inactive municipalities during COVID-19 pandemic.**

| Outcomes | |
|---|---|
| **Active Municipalities** | **Partially active/inactive municipalities**[**] |
| Fulfilment of basic mandates entitled by the central government (S1 File) | Fulfilment of basic mandates entitled by the central government (S1 File) |
| Rapid detection, assessment of cases, and contact tracing | Rapid assessment and detection of cases |
| Free-of-charge provision of COVID-19 and chronic medications to needy people | Free-of-charge provision of COVID-19 medications to needy people |
| Free PCR testing and hospital coverage to economically strained citizens | Provision of fully equipped ambulances |
| Effective management of most cases at home by a trained and qualified team and provision of essential medical utensils (oxygen respirators, oximeters...) | Effective management of most cases at home by a trained and qualified team, and provision of essential medical utensils (oxygen respirators, oximeters...) |
| Provision of fully equipped ambulances | Provision of subsidized medical consultations for COVID-19 cases and contacts in PHCs |
| Designating fully equipped facilities for isolation | |
| Employing statistical models to predict the trend of COVID-19 infection for the coming months and conducting weekly assessment for the current situation | |
| Transforming regular hospital wards into COVID-19 regular and intensive care units | |
| Provision of psychological, social, and occupational therapies to elderly people | |
| Uninterrupted power supply and internet services | |
| Purchase of vaccines to expedite the immunization process in the community | |

*Increased donations* from well-off members in the community, emigrants, and in some cases from NGOs was a major facilitator that enabled all municipalities to thrive during the pandemic despite the scant public financial resources. Besides, *engagement of diversified community actors* (school supervisors, medical teams, Islamic Medical Society, civil society, NGOs...) and task sharing were distinctive features among municipalities with adequate response. On the other hand, municipalities with partial or inadequate response were constrained with limited capabilities and greater mandates. The extent of community engagement was also reflected on the *availability of human resources. For instance*, Khiyam and Amioun municipalities were able to take measures at different levels such as detection, assessment, and management of cases, as well as checking temperature at fixed checkpoints because of the abundance of human resources, unlike municipalities with inadequate or partial response which complained of severe shortage in manpower.

Both Khiyam and Amioun municipalities were described as having *good governance* through their competent municipal members, supportive union of municipalities, adherence to international recommendations and credible healthcare providers. For instance, Khiyam municipality developed their own surveillance database, conducted studies to explore the level of awareness in the community, and used advanced statistical methods to monitor and forecast the dynamics of the virus. On the other side, Michmich and Beyt El Faqs municipalities complained of inactive and unqualified municipal members who were absent during the crisis, and they did not take distinctive measures to halt the pandemic. In addition, they were not relying on credible sources of information to guide their decisions.

*Heightened sense of social support* and empathy among people were among the recurrent themes reiterated by municipalities with adequate and partial/inadequate response. This social support involved a network of family members and friends which helped people cope with the pandemic as described by different stakeholders.

**Table 2. Initiatives taken by municipalities of adequate vs partially adequate/inadequate responses.**

| | Municipality | Infection control and prevention including awareness | Free PCR testing | In-kind and relief assistance | Free provision of COVID-19 medications | Homecare management of COVID-19 cases | Supply of medical equipment and machines | Enforcing of control measures | Overcoming reporting flaws by MOPH | Uninterrupted power supply | Psychologic and social support | Vaccination |
|---|---|---|---|---|---|---|---|---|---|---|---|---|
| **Partially adequate/ inadequate**\*\* | Beyt El-Faqs | × | | × | × | × | × | | | | | |
| | Michmich | × | | | | | | | | | | |
| **Adequate**\* | Amioun | × | × | × | × | × | × | × | | | | × |
| | Khiyam | × | | × | × | × | × | × | × | × | × | |

\* Adeuate: municipalities with active municipal members who took personal initiatives in collaboration with other key actors in the community.

\*\*Partially adequate/inadequate: Municipalities which were completely absent and did not take initiatives beyond the basic mandates, or those with inactive municipal members but where the civil society volunteers overtook the role of the municipality and bridged the gap with their own resources.

Table 3. Enabling factors for municipalities during the pandemic.

| Facilitators | |
|---|---|
| **Municipalities with adequate response** | **Municipalities with partial or inadequate response** |
| Increased donations from well-off citizens and immigrants | Increased donations from well-off citizens and immigrants |
| Heightened sense of social support and empathy in the community | Heightened sense of social support and empathy in the community |
| Engagement of diversified community actors (i.e., local organizations, NGOs, medical societies, school administrators, scouts, community, and religious leaders) | Effective role of PHCs and NGOs |
| Credible sources of information | Credible sources of information |
| Sufficient human resources | |
| Effective role of healthcare systems (i.e., Primary Healthcare Centers, medical laboratories) | |
| Competent municipal members and union of municipalities with action-oriented vision | |
| Employing unique skills (i.e., conducting studies, creating surveillance system, monitoring trends, doing serology testing) | |

*On the effective role of the healthcare system*, both types of municipalities praised the support provided by the primary healthcare centers (PHCs), NGOS, and local laboratories in testing and management of cases. Even in municipalities with inadequate response, the PHCs compensated part of the gap through raising public awareness, doing PCR testing, and supporting preventive measures. Besides, different stakeholders highlighted that healthcare systems capitalized from this pandemic as they got more equipped and recognized.

## C3. Barriers

Barriers consistently raised by both categories of municipalities were: *Economic collapse and poverty, Shortage in resources, Lack of authority and support from the central government, Stigma and lack of awareness, Flaws in MOPH surveillance system, Impeded accessibility to healthcare services, weak role of municipal police, increased mental illnesses, Patronage and Favoritism, Herd immunity and underreporting, and Ignorance of religious leaders* (Table 4) (Fig 2).

*The economic collapse and poverty* were recurrently identified barriers among all municipalities. The pandemic expedited the economic downturn, mounted the rates of poverty, caused massive shut down of businesses, increased the unemployment rate and contributed to the migration of educated people. It was also highlighted that middle-income people and daily-wage earners were most affected. The situation was much more aggravated in rural and distant areas which were already buckling under extreme poverty. On the impact of economic collapse, an official in Michmich municipality revealed that financial hardship was a major factor behind the community resistance saying:

> *"People don't have money to buy a mask, and they use the same mask several days which increases their risk of contracting COVID-19".*

*Shortage in resources* (human, financial, and medical) was illustrated by most of the municipalities at varied level. Beyt El-Faqs and Michmich were complaining of a dearth of human,

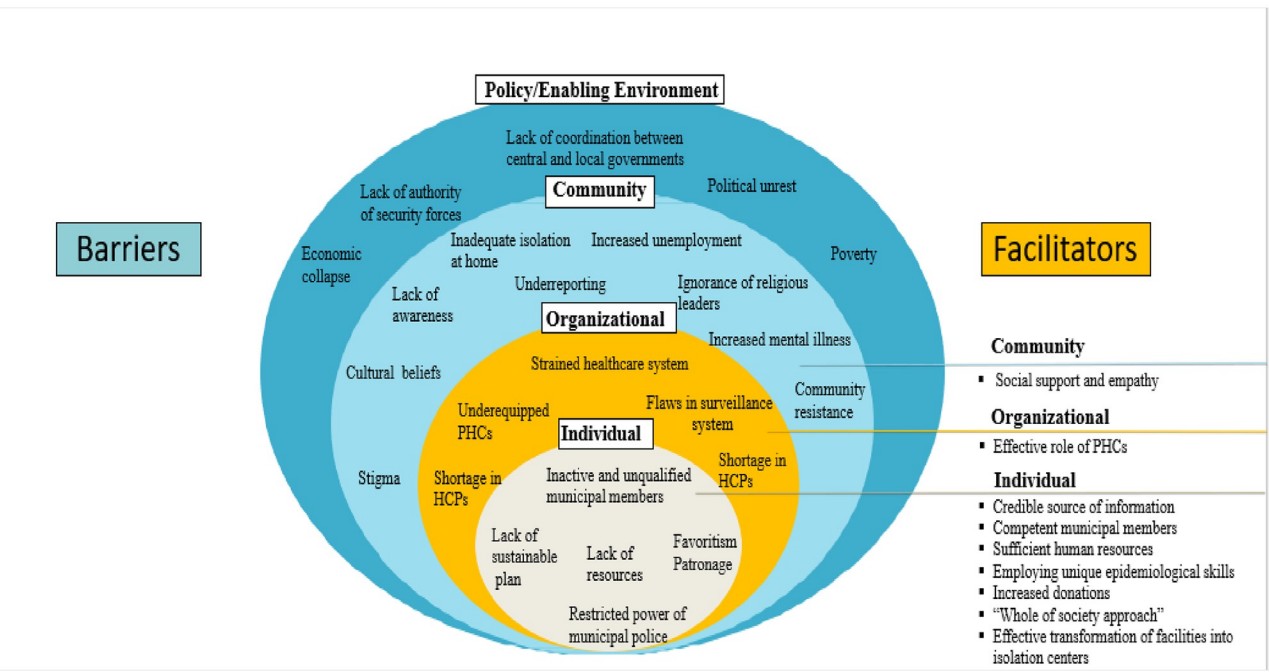

**Fig 2. The ecological framework: Barriers and facilitators affecting municipalities' response to COVID-19.**

financial and other medical and sanitary resources, while Khiyam municipality was facing shortages in financial and essential medical supplies, but not human resources. In this regard, an official in Khiyam municipality and a key informantin Beyt El Faqs municipality, respectively stated:

**Table 4. Barriers facing municipalities during the pandemic.**

| Barriers | |
|---|---|
| **Municipalities with adequate response** | **Municipalities with partial or inadequate response** |
| Economic collapse and poverty | Economic collapse and poverty |
| Scarcity in financial resources and essential utensils (gloves, masks, PPEs) | Scarcity in financial resources, relief supplies, and manpower |
| Lack of authority and support of the central government | Lack of authority and support of the central government |
| Stigma and lack of awareness | Stigma and lack of awareness |
| Flaws in COVID-19 surveillance system of MOPH (i.e., underreporting, delays in reporting PCR results, duplicate names, misclassification of cases per geographical location) | Flaws in COVID-19 surveillance system of MOPH (i.e., underreporting, delays in reporting PCR results) |
| Impeded accessibility to healthcare services (congested governmental and far-distanced hospitals, shortage in experienced healthcare providers, over-priced private hospitals) | Impeded accessibility to healthcare services (congested governmental far-distanced hospitals and negative attitudes towards the quality of care provided, shortage in experienced healthcare providers, over-priced private hospitals, underequipped PHCs) |
| Increased mental illnesses (i.e., dementia, stress, sleep disturbances, anger, depression, psychologic distress) | Increased mental illnesses (i.e., dementia, stress, sleep disturbances, anger, depression, psychologic distress) |
| | Patronage, favoritism, and political interference |
| | Herd immunity and underreporting |
| | Ignorance of some religious leaders who advocate for non-compliance to preventive measure |
| | Weak role of municipal police |

"*Some municipalities are bankrupt and might shut down at any point of time*"

"*The civil society alone cannot fight corona solely without the presence of a complete medical team*".

Both types of municipalities voiced their concerns regarding *the absence of support from the central government* and the lack of authority and prestige of security forces during lockdown. The head of an NGO in Akkar revealed that neither the military nor the municipality police can force people to stop social occasions because of the absence of legislation. On this barrier, a officials in Beyt El-Faqs and Amioun municipalities, respectively stated:

"*The central government is completely absent and the lack of consistency in applying control measures between villages poses a major barrier to the effective implementation of these measures*" "*Although the municipal police force them to close, but when they leave people go inside the shops and shut the doors*"

*Stigma and lack of awareness* were prevailing in both types of municipalities. Lack of awareness, uncontrolled social occasions, and improper home isolation of cases resulted in a vast spread of cases, as iterated by stakeholders. This was coupled by the lack of health consciousness especially among people with low socio-economic level. On stigma, an official and a key informant in Michmich, respectively reported:

"*People consider corona as an enemy that wants to break their social ties and damage their society. . .They believe that God will protect them, and they don't believe in anything else*".

"*Two patients refused that I enter their home because I was wearing PPE. . .so I am publicly exposing them, and this brings them stigma*"

*Increased mental illnesses* (ex. stress, anger, and depression) were a common barrier among both types of municipalities who attributed the cause of these illnesses to the drastic overall situation in Lebanon topped by the pandemic.

*Flaws in the MOPH reporting system* were raised by municipalities with adequate response including Khiyam and Amioun. This was manifested by the lack of accuracy caused by duplicate reporting, wrong names, misclassification of cases per geographical region, and delays in reporting cases which sometimes had recovered from COVID-19 before being accounted. However, this barrier was neither reported by Beyt El Faqs nor Michmich municipalities.

*Access to healthcare services* was a major obstacle faced by municipalities with adequate and inadequate/partial response. The healthcare system collapsed, governmental hospitals became congested, while only few overpriced private hospitals opened COVID-19 wards. In addition, rural areas were further strained by the far-distanced hospitals and shortage in experienced healthcare professionals. In this regard, an official in Beyt El Faqs municipality, and a key informant Khiyam municipality, respectively iterated:

"*There are no available beds in governmental hospitals, private hospitals are overpriced, and all hospitals are far-distanced. We also find difficulty in transporting patients by equipped ambulances*".

"*I can't believe that only 67 out of 314 hospitals are assigned to receive COVID-19 patients at the time when Italy employed 90% of its private hospitals during the pandemic*"

*Patronage and favoritism* were prevailing in municipalities with inadequate/partial response while political interference was also mentioned by municipalities with adequate response. On the political interference, some municipalities were unable to designate an isolation center due to political interference. In this regard, a key informant in Khiyam municipality complained:

*"Unfortunately, the municipalities are still buckling under political influence which are infringing on the developmental mission of municipalities and politicizing them".*

Michmich and Khiyam municipalities referred to herd immunity as a result of the uncontrolled spread of the virus coupled with the lack of compliance to preventive measures. In Michmich, most of the residents contracted COVID-19, and large death tolls were recorded, but neither the cases nor the deaths were confirmed by or reported to MOPH. Underreporting in Michmich and Beyt El Faqs was attributed to stigma, conspiracy theories, and financial hardships that impeded people from doing PCR testingas illustrated by stakeholders in Michmich municipality:

*"95% of cases in Michmich are not reported and we are relying on herd immunity"*

*"I bet if the government does PCR, they will detect more than 20000 cases in Michmich"*

Ignorance of religious leaders was mentioned by municipalities with inadequate/partial response. A stakeholder in Beyt El-Faqs reported that some religious leaders were skeptical regarding the presence of COVID-19, and they were inciting people to disbelieve in it. He added that people visiting the mosques were not putting masks and were not abiding to any preventive measure.

## D. Discussion

The aim of this study was to document the spectrum of preparedness and reactivity displayed by municipalities in Lebanon during the COVID-19 pandemic (2020–21), and to pinpoint the facilitators and the barriers that faced them. One of the key observations was the discrepancy in the levels of preparedness and reactivity of municipalities as well as the graduation in the threats and facilitators which were also recognized across distinct geographical location. This is first and foremost attributable to the absence of a clear-cut, well-defined and well-implemented central plan to address the epidemic in a uniform and equitable way across the Republic.

Numerous facilitators enabled the municipalities to thrive during the pandemic. It had been clearly recognized from previous crises that traditional and sole response of local governments was ineffective in preventing and controlling pandemics. A holistic preparedness approach requires active cooperation between the state, local governments, and the neighborhood communities. Successful municipalities in Lebanon integrated a community-based response to help them bridge the gaps that were shaped by the sparse resources and the lack of support from the government. The role of the civil society was outstanding in almost all municipalities, and in some cases, they overtook the role of the local government during the pandemic. In that regard, Taiwan can be considered as a successful case-study as it adopted the "whole-of-society" approach which calls for resource sharing, trust building, and engaging the community in decision making [10]. Beside the availability of human resources, municipalities capitalized on fundraising and donations collected from the well-off members in the community at home and abroad. Without these donations, the municipalities would have

been paralyzed and crippled given the severe financial challenges from which they are suffering.

Another key aspect of successful municipalities was having good governance manifested by the presence of competent and qualified municipal members who promptly responded to the crisis and effectively utilized their resources. Besides, adherence to credible sources of information was a crucial element which enabled municipalities to successfully manage during the crisis unlike municipalities which were relying on social media platforms. The literature indicated that misleading posts on social media platforms were way more recurrent and recognized compared to accurate and credible posts, and that over one-quarter of YouTube videos which were viewed by thousands of people contained misleading information [11, 12]. This calls upon public health agencies to harness the power of these channels to disseminate accurate and timely information to the public.

On the healthcare level, experienced healthcare providers are precious assets in the community, not only for caring of patients, but also for their indispensable role in controlling outbreaks [13]. In addition, the primary healthcare system played a significant role during the pandemic, and its role was more pronounced in rural and deprived settings despite being under-resourced and underequipped. This highlights the urgent need to strengthen the role of PHCs and to upgrade the quality of services provided, since PHCs can be the mainstay in the face of emergency situations especially in low-resource settings. In universal public health system, PHCs are at the core of ministerial agendas since it is the only way to provide accessible, affordable, sustainable and comprehensive medical care that aims to reduce health inequities in the community [14].

Social cohesion in crisis is a fundamental element for stability. In this study, several municipalities provided psychological support to the residents, especially for elderly who were severely impacted by the crisis due to isolation and the migration of their children. In addition, it was stressed by several municipalities that there was a heightened sense of social support and empathy in the community. This aligns with other studies that showed that social support and the level of caring among family members increased during the pandemic [15].

Regarding the barriers, it is important to emphasize that COVID-19 pandemic in Lebanon is not just a health crisis, it is a complex interplay of several other pandemics, including political, economic, social and health crises. Although the majority of the barriers were common among all municipalities, not all municipalities were able to overcome these barriers and respond effectively during the crisis.

At the individual level, many municipalities had incompetent people assuming municipal positions, with some municipal council members in rural areas were illiterate and lack essential health-related knowledge. This dilemma stemmed from the power-sharing political system that thrives on patronage, favoritism, and sectarian and political divisions [5].

Although Lebanon had faced several other epidemics before COVID-19, yet the central and local governments did not scale up their capabilities and designate resources for future crisis. Thus, the level of preparedness was inappropriate causing delay in response. In contrast, China, Taiwan, and South Korea were able to halt the progression of the pandemic through early decisive actions and adequate preparedness level [16]. However, the sustainability of the preparedness plan requires huge resources and capacities which were lacking in this context. All municipalities complained of scant financial resources and the delays in paying their receivables, which paralyzed them and hindered their efforts. However, rural remote municipalities were way more affected than the ones proximal to the capital Beirut, because they were suffering from sparse human, financial, and relief assistance. The case of Lebanon resembles that of Yemen which lacks the technical capacity and prioritization of resources because of the fragmented and fragile authority [17]. However, it was not only about the lack of prioritization

of resources, but also the inequitable allocation of resources and support from the government. These findings align with a systematic review conducted in Sub-Saharan Africa which pointed out that politicized allocation of health resources and skills on health services provision by local governments. The allocation of resources and skills were characterized by consistent state and political interferences [18]. Given that the healthcare system in Lebanon is predominantly private and overpriced, this increased the health inequity gap in the society especially that governmental hospitals were overwhelmed, and they were the first to respond to this public health emergency. Poorer and vulnerable population were left grappling to find a bed in the fully occupied hospitals because they cannot afford private ones. The situation is much worse as we go further away from the capital, where hospitals are far-distanced and underequipped. Besides, the quality of care provided is questionable given the severe shortage in experienced nurses and healthcare providers [19]. In situation where most cases do not need to be hospitalized, the role of municipalities in managing cases at home and providing them with the essential medical supplies (under the supervision of a qualified team is pivotal and can spare a lot of people all sorts of burden related to hospital admissions.

On the community level, lack of awareness manifested by the prevalence of fake news, conspiracy theories and low literacy rate constituted one of the biggest hurdles that contributed to the lack of adherence. Besides, cultural beliefs and the collectivist nature of the Lebanese society resulted in massive spread of the virus and impeded municipalities from successfully enforcing control measures [20]. Strongly held religious beliefs that "we rely on our faith and God will protect us" were also a major barrier to limited testing and lack of adherence to preventive measures. Alongside, the ignorance of some religious leaders in health issues worsened the scenario through urging people to disbelieve in this pandemic and condemn the usefulness of the precautionary measures. Thus, informing religious leaders and recruiting them is important to foster better public health response [21].

Underreporting of cases was clearly recognized from the IMPACT platform for municipalities where the number of cases reported on this platform is much lower than the numbers mentioned by the interviewed stakeholders. This discrepancy originated from the refusal of people to do PCR testing and to get discriminated and stigmatized. Stigmatization was ignited since COVID-19 is a novel virus and infodemic were invading the media platforms [22]. Consistent with the Ebola pandemic, stigmatization shaped by discrimination, social isolation, and prejudice not only hampered the opportunities to halt the spread of the disease but resulted in augmented transmission and higher mortality [23, 24]. Stigma is the hidden threat to optimal COVID-19 control, as it can be associated with refusal to seek medical care, underreporting, and delayed representation of cases which had drastic impact at the individual and public health levels [25]. Although the government launched several awareness campaigns to educate people, they were not taken seriously because barriers along with the frail social contract between the government and the public. Such barriers require effective strategies of risk communication to instruct, inform, and provide proactive public health response that can prevent fake infodemics and foster better response from the community [26].

On the policy level, lack of coordination between the central and local governments hindered the response of municipalities in combatting the pandemic. The decentralized and fragmented decisions in the absence of good governance during the epidemic resulted in massive and vast spread of the disease [27]. Besides, the lack of prestige and authority of security forces was associated with chaos, uncontrolled spread of the virus, and inadequate decisions. Thus, when there is no authority from the state, laws cannot be enforced, and people cannot be blamed for not adhering.

In addition, several flaws and shortcomings in the governmental surveillance system were identified (inaccurate reporting of cases, misclassification of cases per geographical location,

delayed reporting of confirmed or recovered cases, and counting duplicates). Besides, some municipalities referred to the absence of coordination between private laboratories and the MOPH surveillance system causing underreporting of cases. Studies have shown that insufficient testing and tracing of contacts cause underreporting, and that mortality rates provide a better indicator to quantify the burden of COVID-19 infection in such cases [28]. Studies have shown that low- and middle-income countries tend to have weaker health protection systems and public health infrastructure which impede them from having timely and accurate notification system [29]. In addition, the surveillance system in these settings is more rudimentary and lack analytical strength [30]. Since the surveillance system is the eyes of the health system, it is crucial to invest in health protection infrastructure to guide policymaker's decisions to mitigate the burden of infection in Lebanon.

This pandemic has ravaged the already multiple-stricken country. Lebanon is witnessing the worst economic crisis manifested by the devaluation of currency, massive shutdowns of businesses, unprecedented rates of unemployment rate, and mounting poverty rates. The stringent governmental control decisions during the pandemic were not coupled by any contingency plan to mitigate the impact of this crisis on the population and it did not balance between economy and public health. A lot of people slipped into poverty and the real GDP growth decreased by 20% in 2020 [31]. This had yielded strong community resistance, lack of adherence, and devastating health outcomes. At the same time, the government did not intervene to mitigate the impact of the economic crisis on the population compared to other countries. In contrary, governments around the globe took fiscal, financial, and monetary policies to alleviate the impact of control measures on the economy while ensuring the public welfare [32].

The mental health of people was undermined because of the military and political conflicts, and economic downturn that afflicted the country since decades [33]. Besides, the mental well-being of people had been compromised as a result of the pandemic which imposed social distancing, isolation, lack of access to essential supplies, and stigma [34]. Alongside, the detrimental impacts of COVID-19 on health and the death toll statistics instigated fear among people and compromised their mental health [35]. This was clearly articulated by several municipalities that recognized a surge in mental health illnesses during the pandemic and was manifested by stockpiling of psychotropic medications. The findings of this study align with a recent study conducted in Lebanon which showed that there is strong correlation between depression, anxiety, stress among the Lebanese population during COVID-19 pandemic [36]. In addition, mental health illnesses are associated with emotional responses, unhealthy behaviors, and non-compliance to health preventive decisions among patients and the public. This necessitates prompt interventions from health authorities and NGOs to provide psychological and mental support to affected people and to enhance their coping with this pandemic [37].

## E. Conclusions

The COVID-19 pandemic eroded the resilience of the already crippled central government, thus offering an opportunity for municipalities to innovate and bridge the gap in facing this one more difficulty. Municipalities were able to overcome the sparse resources and the absence of governmental support through good governance, community engagement in decision making, and resource sharing. During pandemics one size does not fit all and having a national response and preparedness plan that does not take into consideration the capacity gaps and the widened inequities among regions cannot thrive. Several loopholes were recognized at the individual, organizational, community, and policy levels and strategic roadmap was developed to address these multi-level barriers (S6 File).

## Supporting information

**S1 Fig. Epidemic curve of covid-19 in Lebanon (2020–2021).**
(TIF)

**S1 File. Basic mandates of municipalities as requested by the government.**
(DOCX)

**S2 File. Facilitators, barriers, and outcomes Beyt El Faqs municipality.**
(DOCX)

**S3 File. Facilitators, barriers, and outcomes of Michmich municipality.**
(DOCX)

**S4 File. Facilitators, barriers, and outcomes of Amioun municipality.**
(DOCX)

**S5 File. Facilitators, barriers, and outcomes of Khiyam municipality.**
(DOCX)

**S6 File. Roadmap to enhance the response of municipalities during crisis based on the results of the ecological framework.**
(PDF)

## Acknowledgments

This research was conducted as part of the graduation requirements of JEM with the MPH degree at the American University of Beirut which hosted the activities free-of-charge. The authors wish to thank Dr. Aline Germani for her critical review of an earlier version of this manuscript.

## Author Contributions

**Conceptualization:** Jida M. Al-Mulki, Mahmoud H. Hassoun, Salim M. Adib.

**Data curation:** Jida M. Al-Mulki.

**Formal analysis:** Jida M. Al-Mulki.

**Investigation:** Jida M. Al-Mulki, Mahmoud H. Hassoun.

**Methodology:** Jida M. Al-Mulki, Salim M. Adib.

**Supervision:** Salim M. Adib.

**Validation:** Salim M. Adib.

**Writing – original draft:** Jida M. Al-Mulki, Salim M. Adib.

**Writing – review & editing:** Salim M. Adib.

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
