## [Decision Letter · Decision Letter 0]

25 Nov 2021

PONE-D-21-23335EPIDEMICS AND LOCAL GOVERNMENTS IN STRUGGLING NATIONS:  COVID-19 IN LEBANONPLOS ONE

Dear Dr. Jida Mulki,

Thank you for submitting your manuscript to PLOS ONE. After careful consideration, we feel that it has merit but does not fully meet PLOS ONE’s publication criteria as it currently stands. Therefore, we invite you to submit a revised version of the manuscript that addresses the points raised during the review process.

We look forward to receiving your revised manuscript.

Kind regards,

Rogis Baker, Ph.D

Academic Editor

PLOS ONE

Journal Requirements:

a) Did participants provide their written or verbal informed consent to participate in this study? 

3. When reporting the results of qualitative research, we suggest consulting the COREQ guidelines  or other relevant checklists listed by the Equator Network, such as the SRQR, to ensure complete reporting (http://journals.plos.org/plosone/s/submission-guidelines#loc-qualitative-research). Moreover, please provide the interview guide used as a Supplementary File.

5. We note that Figure 1 in your submission contain map images which may be copyrighted. All PLOS content is published under the Creative Commons Attribution License (CC BY 4.0), which means that the manuscript, images, and Supporting Information files will be freely available online, and any third party is permitted to access, download, copy, distribute, and use these materials in any way, even commercially, with proper attribution. For these reasons, we cannot publish previously copyrighted maps or satellite images created using proprietary data, such as Google software (Google Maps, Street View, and Earth). For more information, see our copyright guidelines: http://journals.plos.org/plosone/s/licenses-and-copyright.

a) You may seek permission from the original copyright holder of Figure 1 to publish the content specifically under the CC BY 4.0 license.  

Natural Earth (public domain): http://www.naturalearthdata.com/.

6. We note you have included a table to which you do not refer in the text of your manuscript. Please ensure that you refer to Table 2 in your text; if accepted, production will need this reference to link the reader to the Table.

7. Please include your tables as part of your main manuscript and remove the individual files. Please note that supplementary tables (should remain/ be uploaded) as separate "supporting information" files.

Reviewers' comments:

Reviewer's Responses to Questions

**Comments to the Author**

1. Is the manuscript technically sound, and do the data support the conclusions?

Reviewer #1: Yes

Reviewer #2: Yes

2. Has the statistical analysis been performed appropriately and rigorously? 

Reviewer #1: N/A

Reviewer #2: N/A

3. Have the authors made all data underlying the findings in their manuscript fully available?

Reviewer #1: Yes

Reviewer #2: Yes

4. Is the manuscript presented in an intelligible fashion and written in standard English?

Reviewer #1: Yes

Reviewer #2: Yes

5. Review Comments to the Author

Reviewer #1: This manuscript presents the findings of very timely research that can help local authorities around the world (and not just in countries facing similar challenges to Lebanon) in dealing with current and future pandemics. Overall, I thought the document was well-written and easy to follow. There were, however, a few points for the authors to consider. I will start with the major point to address:

1. My main concern relates to preserving the identity of respondents. The authors state in their “ethical considerations” section that “potential participants were informed that names will not be disclosed in any report”. However, I suspect that the identity of some of the respondents could be revealed from the description of their role. For example, “the Mayor of Khiyam, or the Mayor of Michmich”. I would strongly encourage respondents to either anonymise the names of all the municipalities or describe roles in more generic terms, for example “Official in municipality”. Alternatively, the authors could explain why there is no risk of the identity of respondents being revealed.

2. Other points for the authors to think about:

a. The time period when the data were collected (i.e. interviews were conducted) is not clear. It would be helpful to know whether this was in the very early stages of the pandemic, or later on. It would also be helpful for the reader to know the duration during which data were collected, because I imagine some of the municipalities may have moved between the two categories (i.e. their response to the pandemic may have improved or become worse).

b. The research was carried out in nine municipalities, however the authors restrict their analysis to four municipalities. It would be helpful to explain the rationale for this.

c. One of the key discussion points relates to the proximity of municipalities from the capital. However, this is not described in sufficient detail in the results. It would be helpful to know, for example, whether both of the municipalities with adequate response were closer to the capital than those with inadequate response. The map may be helpful in this respect if it showed where capital was located on the map.

d. Some of the statements in your discussion could have benefitted from references. For example, the entire discussion about the inequitable distribution of resources (on page 26) is not supported by existing literature.

Finally, here are some miscellaneous issues for the authors to consider:

a. Describing the central government as “eternally” broke is in my view too broad a statement, and a more specific time-frame will need to be provided.

b. Similarly, the lack of awareness of religious leaders could perhaps be made more specific. I.e. is it ignorance of health issues rather than ignorance in general?

Reviewer #2: The manuscript covers an interesting and important aspect of the pandemic. Below please find a few suggestions to further strengthen the manuscript.

1) Abstract (page 3) Please quantify "most municipalities" for example in %

2) (page 8) Please elaborate on "the felt presence to the virtually absent". It is difficult to understanding the meaning

3) (page 8) Please justify why SEM was used rather than other frameworks. And outline advantages and disadvantage

4) Methods (p1ge 11) Could you provide the IRB number?

5) The results and discussion sections are lengthy, making it challenging for the reader to focus on the core messages. Please try to modify/shorten the text to make it more coherent and succinct

6) Please carefully check for typos and correct grammar e.g. (page 8 SEM has been adapted

6. PLOS authors have the option to publish the peer review history of their article (what does this mean?). If published, this will include your full peer review and any attached files.

Reviewer #1: No

Reviewer #2: No

---

## [Author Response · Author response to Decision Letter 0]

11 Dec 2021

Academic Editor:

The manuscript is adjusted according to PlOS One guidelines.

a) Did participants provide their written or verbal informed consent to participate in this study? 

The ethics statement was amended as requested.

3. 3. When reporting the results of qualitative research, we suggest consulting the COREQ guidelines or other relevant checklists listed by the Equator Network, such as the SRQR, to ensure complete reporting

The manuscript was checked against the COREQ guideline and complete reporting was ensured.

4. We note that you have stated that you will provide repository information for your data at acceptance. Should your manuscript be accepted for publication, we will hold it until you provide the relevant accession numbers or DOIs necessary to access your data.

Reserved DOI: 10.17632/k4jp3xscg2.1 

5. We note that Figure 1 in your submission contain map images which may be copyrighted. All PLOS content is published under the Creative Commons Attribution License (CC BY 4.0)

Permission to use the map is included in “other” supplement.

6. We note you have included a table to which you do not refer in the text of your manuscript. Please ensure that you refer to Table 2 in your text;

Table 2 was cited

7. Please include your tables as part of your main manuscript and remove the individual files. Please note that supplementary tables (should remain/ be uploaded) as separate "supporting information" files.

Tables were added to the manuscript.

8. Please include captions for your Supporting Information files at the end of your manuscript, and update any in-text citations to match accordingly.

Captions were added.

 Reviewer 1:

1. I would strongly encourage respondents to either anonymise the names of all the municipalities or describe roles in more generic terms, for example “Official in municipality”.

All identifiers were replaced with generic terms (eg, officials, stakeholders, key informants)

2. a. The time period when the data were collected (i.e. interviews were conducted) is not clear

The time period of the conducted interviews was added in the study procedures section. 

b. The research was carried out in nine municipalities, however the authors restrict their analysis to four municipalities. It would be helpful to explain the rationale for this. 

The rational for including 4 municipalities in this paper is to keep it short, concise, and readable (added this to the plan of analysis) 

c. One of the key discussion points relates to the proximity of municipalities from the capital. However, this is not described in sufficient detail in the results. It would be helpful to know, for example, whether both of the municipalities with adequate response were closer to the capital than those with inadequate response. The map may be helpful in this respect if it showed where capital was located on the map.

The response of the municipalities to the capital in regard to their response is more pronounced for the 9 municipalities, with most of the proximal municipalities responded better to the pandemic due to the availability of resources and better healthcare systems. However, I removed the sentence that relates proximity to the response because on of the selected municipalities which responded well to the pandemic is not proximal to the capital (khiyam).

d. Some of the statements in your discussion could have benefitted from references. For example, the entire discussion about the inequitable distribution of resources (on page 26) is not supported by existing literature.

References are added

Finally, here are some miscellaneous issues for the authors to consider:

a. “eternally” was removed

b. The ignorance of religious leaders was modified to “ignorance in health issues”.

Reviewer 2: 

1) I opted not to include percentages because in this qualitative study, we are not interested in quantification, rather we want to get in depth insights on the experiences of various municipalities during the pandemic. If I put percentage in the abstract, then I must add percentages to all the results.

2) (page 8) Please elaborate on "the felt presence to the virtually absent". It is difficult to understanding the meaning. Some municipalities were distinguished by the activities they undertook, thus making their presence noticed by people, while other municipalities get unnoticed because they were not active during the pandemic 

3) (page 8) Please justify why SEM was used rather than other frameworks. And outline advantages and disadvantage

I used SEM because it is an ideal tool to address complex public health issues, as it allows deep understanding of the dynamic interplay between personal, interpersonal, organization, community and policy levels. In other words, we can see the multi-level barriers that impeded municipalities from responding well to the pandemic and design interventions at different levels (individual, community, policy…) to overcome these barriers. However, one of the major imitations of SEM is that the programs can be expensive to implement. In addition, it requires close coordination between individuals and groups for effective implementation.

4) Methods (p1ge 11) Could you provide the IRB number? 

IRB number: SBS-2020-0428

5) The results and discussion sections are lengthy, making it challenging for the reader to focus on the core messages. Please try to modify/shorten the text to make it more coherent and succinct

I shortened the results and discussion as much I can, but this study has a lot of themes and subthemes that cannot be omitted or overlooked.

6) Please carefully check for typos and correct grammar e.g. (page 8 SEM has been adapted

 The paper was spell checked and corrected.

---

## [Editor Report · Decision Letter 1]

16 Dec 2021

EPIDEMICS AND LOCAL GOVERNMENTS IN STRUGGLING NATIONS:  COVID-19 IN LEBANON

PONE-D-21-23335R1

Dear Dr. Jida mulki,

We’re pleased to inform you that your manuscript has been judged scientifically suitable for publication and will be formally accepted for publication once it meets all outstanding technical requirements.

Kind regards,

Rogis Baker, Ph.D

Academic Editor

PLOS ONE
---

## [Editor Report · Acceptance letter]

6 Jan 2022

PONE-D-21-23335R1 

Epidemics and local governments in struggling nations:
COVID-19 in Lebanon 

Dear Dr. Al-Mulki:

I'm pleased to inform you that your manuscript has been deemed suitable for publication in PLOS ONE. Congratulations! Your manuscript is now with our production department. 

Kind regards, 

on behalf of

Dr. Rogis Baker 

Academic Editor

PLOS ONE